# Impact of Public Environmental Concerns on the Digital Transformation of Heavily Polluting Enterprises

**DOI:** 10.3390/ijerph20010203

**Published:** 2022-12-23

**Authors:** Youmeng Wu, Hao Sun, Hongliang Sun, Chi Xie

**Affiliations:** 1School of Business, Anhui University of Technology, Maanshan 243002, China; 2School of Business, Hunan Normal University, Changsha 410081, China

**Keywords:** public environmental concerns, heavily polluting enterprises, digital transformation, China

## Abstract

China is currently facing the arduous tasks of energy conservation, emission reduction and structural transformation, making it of great significance to study the digital transformation of heavily polluting enterprises. As an important informal regulatory system, public environmental concerns affect corporate environmental behavior by increasing external environmental pressure. This study uses the data of listed companies in China’s heavily polluting industries from 2012 to 2020 and Baidu Index data to analyze how public environmental concerns affect the digital transformation of heavily polluting enterprises. This study finds that public environmental concerns can significantly promote the digital transformation of heavily polluting enterprises. For non-state-owned, green image and high-tech enterprises, the impact is even more obvious. Furthermore, based on the structural and hierarchical perspective of enterprise digital transformation, we find that public environmental concerns significantly promote digital technology application. This study puts forward some suggestions for government departments to formulate environmental protection regulations, enterprises to fulfill their green responsibilities and the public to participate in environmental governance.

## 1. Introduction

Since its reform and opening up, China’s economy has maintained high-speed growth, but the pressure placed on the ecological environment is increasing daily, resulting in environmental incidents such as those related to water, air and food pollution [1]. As an important informal regulatory system, public concern plays an important role in the process of environmental governance [2]. With the improvement of public awareness of environmental protection and the popularity of online media such as social media, short video apps and search engines, the public has more channels through which to understand, disseminate and share environmental information and express its views on environmental governance [3]. The resulting public opinion affects not only the environmental governance decisions of the government but also the environmental behavior of enterprises [4]. At the same time, to solve the problem of a lack of public participation, in recent years, the government has begun to consciously strengthen environmental information disclosure and actively guided the public to express environmental demands in a reasonable and legal manner [5]. Previous studies have shown that public environmental concerns can reduce the degree of information asymmetry between the government and enterprises, restrain government–enterprise collusion [6] and help reduce pollutant emissions or improve the pollution control level of enterprises [7]. In addition, public environmental concern can not only increase the return of corporate environmental protection but also magnify the cost of corporate environmental pollution and constantly reduce the operating performance of polluting enterprises [8]. As an important part of pollution discharge, heavily polluting enterprises have strong negative externalities in the production process, and thus, it is urgent to find a green transformation method for polluting enterprises, especially heavily polluting enterprises.

With the rapid development of emerging information technologies such as artificial intelligence, big data and blockchain, digital technology has become a powerful driving force for high-quality economic development, playing a key role in enterprise development [9,10]. As the main micro body of the high-quality development of the national economy, digital transformation is the only way for enterprises to comply with the law of the development of the times [11]. The digital transformation of enterprises can promote their green innovation, such as process innovation, business model innovation, and product innovation [12,13,14,15]. Moreover, the green innovation behavior of enterprises can improve their ability to design green products and services, thereby reducing their emission of harmful pollutants and consumption of natural resources [16]. In addition, enterprise digital transformation helps enterprises collect operational data in real time, facilitate energy management and the predictive maintenance of equipment and reduce energy consumption and carbon emissions [17]. Especially for heavily polluting enterprises, digital transformation can help them improve the level of environmental governance and reduce pollutant emissions to regain the public’s “vote”, or approval, in the commodity and capital markets and establish a new competitive advantage.

However, there are few studies on the impact of public environmental concerns on the digital transformation of heavily polluting enterprises. Can public environmental concerns promote the digital transformation of heavily polluting enterprises? Are the impacts different? For this reason, this study takes listed companies in China’s heavily polluting industries from 2012 to 2020 as the research sample and conducts screening and processing on the samples using negative binomial regression and other models to empirically test the impact of public environmental concerns on the digital transformation of heavily polluting enterprises. The marginal contributions of this study are as follows. (1) From the perspective of public environmental participation, this work analyzes the impact of public environmental concerns on the digital transformation of heavily polluting enterprises, which enriches the research on the impact of informal environmental regulations on enterprise environmental behavior. (2) Based on enterprise characteristics, such as enterprise property rights, green image and technology level, this work analyzes the heterogeneous impact of public environmental concerns on the digital transformation of heavily polluting enterprises, providing a logical explanation for such transformation.

## 2. Theoretical Analysis

In recent years, with the development of digital technology represented by “artificial intelligence”, “blockchain”, “cloud computing” and “big data”, more and more enterprises are joining the digital wave, which brings new opportunities for enterprises to implement energy saving and emission reduction [18]. Digital transformation refers to the upgrading of organizational structures and business models through information technology. The technological progress and digital communication technology applications embedded in digital transformation can not only improve the efficiency of enterprise production factors such as utilization and energy use efficiency [19,20], but also promote the fine management of material input, product manufacturing and sales processes through the automation upgrade of production processes, prompting enterprises to accurately control the production process while monitoring the energy consumption of each link in real time, reducing energy consumption [21,22,23]. This will reduce the rate of energy consumption, reduce waste in production and generate positive feedback on energy saving and emission reduction. The above analysis shows that digital transformation can promote energy saving and emission reduction in enterprises. However, whether enterprises are motivated to engage in environmentally friendly behavior in various ways depends largely on the internal and external environment, in which the pressure of environmental regulations on enterprises plays an important role [24,25,26,27].

The ethics formed by public environmental concern acts as a type of informal regulation that affects the behavior of enterprises, especially those in heavily polluting industries that face greater pressure to pay attention to the environment and more urgently respond to public environmental concerns [2,28,29]. In general, in areas with more public environmental demands, the government enforces environmental laws and regulations more strictly, and the environmental requirements for enterprises are thus higher [18]. In addition, when an enterprise violates environmental ethics, the public can form its own strong opinion and thus affect the business performance of the enterprise and promote the environmental behavior of the enterprise through the media’s public reports on its pollution level or environmental governance level [29]. In the capital market, environmental information disclosure changes the public’s investment behavior and consumption tendency, that is, “voting with money” [30]. Public environmental concern has undoubtedly increased the cost of environmental protection for enterprises, forcing them to save energy and reduce emissions. Compared with direct environmental protection investment, enterprise digital transformation has the potential to bring with it long-term benefits [31,32]. First, the digital transformation of enterprises can give play to the “resource effect”. By improving the financing capacity of enterprises, digital transformation can ease the financing constraints of enterprises to promote their environmental protection practices and green technology innovation [33]. Second, the digital transformation of enterprises can give play to the “governance effect” and improve the environmental protection management level of enterprises by strengthening external supervision and mitigating internal agency conflicts [30,34]. Finally, the digital transformation of enterprises can have a “multiplier effect” in that it can stimulate the growth potential of enterprises by optimizing their allocation of internal and external resources [35] to provide endogenous power for green activities with long-term benefits. Based on the above analysis, this study proposes research Hypothesis 1:

**Hypothesis** **1.***Public environmental concerns have significantly promoted the digital transformation of heavily polluting enterprises*.

As the leading force of the national economy, the governance system of state-owned enterprises is fundamentally different from that of private enterprises in China [36]. In terms of corporate environmental governance, some people believe that state-owned enterprises perform well in terms of environmental performance [37,38], whereas others believe that the environmental performance of state-owned enterprises is worse than that of non-state-owned enterprises [39]. Various theories aim to explain the different degrees of environmental performance of different ownership types, including political connections [39], the resource-based view (RBV) [40] and multitask theory [41]. This study complements relevant theories by emphasizing the effectiveness of enterprise digital transformation in enterprise environmental governance. Under pressure from public environmental concerns that are not mandatory for companies to upgrade their environmental behavior, state-owned enterprises may have less enthusiasm to respond. On the one hand, state-owned enterprises consider the government the controlling shareholder and, thus have closer ties with it, which makes it easier for them to collude with the government and evade environmental responsibilities [42]. On the other hand, the administrative level of some state-owned enterprises can be even higher than that of local government officials, which makes it difficult for local governments to supervise these state-owned enterprises [43,44]. In addition, there may be differences in the way state-owned and private enterprises achieve environmental performance. In consideration of economic benefits, private enterprises are more inclined to adopt technology transformation measures that can bring about long-term benefits and enhance enterprise competitiveness, such as the digital transformation of enterprises, whereas state-owned enterprises are more inclined to achieve environmental performance directly by increasing environmental input to meet the needs of their shareholders, that is, the interests of the government [41,45]. Based on the above analysis, this study proposes research Hypothesis 2:

**Hypothesis** **2.***Compared with state-owned enterprises, public environmental concerns have a more obvious role in promoting the digital transformation of non-state-owned enterprises*.

Enterprises with a green image often pay more attention to environmental governance to maintain their green image and market competitiveness. In addition, the green image of the enterprise also causes the public to develop a positive attitude toward environmental governance [45], which makes it easy for enterprises to gain consumers’ trust. For example, the public tends to buy products with green labels in the commodity market [46], making it easier for enterprises to obtain the “vote”, or approval, of the public in the investment and stock markets [47]. Under this condition, enterprises have stronger power to carry out green transformation and upgrading to better meet the public’s demand for green products. For an enterprise, having a green logo means that it has a more advanced concept of green development and a clearer understanding of green environmental protection [48]. Relatively speaking, such enterprises have strong receptivity to digital transformation and are willing to actively embrace the era of digital transformation. Based on this aspect, this study proposes research Hypothesis 3:

**Hypothesis** **3.**
*Corporate green image plays a positive role in regulating the impact of public environmental concerns on the digital transformation of heavily polluting enterprises.*


Technological innovation is an inexhaustible driving force leading national development, the core of enterprise digital transformation and a new engine with which to promote enterprise development [49]. For enterprises in different industries, even among enterprises of the same type, there is a gap between high and low technical levels, and there are also great differences in the cognition and understanding of digital transformation. For enterprises with a weak technology level, there may be a large cognitive gap in the actual application scenario of digital technology, which directly affects their decision about digital transformation. Enterprises with a high technology level have a deep understanding of digital transformation and, thus attach great importance to it; they are more willing to run digital transformation through all aspects of enterprise production, operation, management, sales, etc. [9]. At the same time, the marginal increase in the R&D cost of such enterprises is often lower than that of enterprises with low technology levels, so they have a strong internal motivation to carry out digital transformation. Based on the above analysis, this study proposes research Hypothesis 4:

**Hypothesis** **4.***A technological foundation plays a positive role in regulating the impact of public environmental concerns on the digital transformation of heavily polluting enterprises*.

## 3. Data and Methodology

### 3.1. Variables and Data

#### 3.1.1. Dependent Variable

Our research focuses on the impact of public environmental concerns on corporate digital transformation, which has been shown to be an important means for companies to implement environmentally friendly behaviors [50,51]. In China, in 2021, the State Council issued the “Action Plan to Achieve Carbon Peak by 2030”, which calls for the in-depth implementation of green manufacturing projects, the vigorous implementation of green design and the improvement of green manufacturing systems, further clarifying the importance of promoting the integration and development of digitalization, intelligence and greening in the industrial sector. According to the World Economic Forum, the carbon emissions reduced by industries benefiting from information and communication technology (ICT) will reach 12.1 billion tons by 2030. Thus, it is clear that digital technology has become an emerging dividend to lead enterprises to green manufacturing, which provides some realistic basis for the selection of our dependent variables.

The quantitative measurement of enterprise digital transformation is a frontier issue in both academia and practice. Enterprise digitization does not involve simply the digitization of enterprise data but rather the promotion of the digitization of enterprise production materials and processes with the help of cutting-edge digital technology and hardware systems. Based on this, this study draws on the processing method of Wu et al. [52] and identifies and extracted keywords related to digital transformation based on the text data of the annual reports of listed companies. Moreover, starting from the process of enterprise digital transformation, the characteristic words of enterprise digital transformation are divided into two categories: digital transformation technology types and digital technology applications. Specifically, the underlying technology architecture of digital transformation includes four mainstream technology directions: artificial intelligence, blockchain, cloud computing and big data. At the practical level of technology application, feature words include mainly the scene application of digital business. Through the induction and classification of feature words in digital transformation, five sets of feature words are finally formed. The basis of induction and detailed classification are shown in the notes (Table A1). Furthermore, based on the text data of enterprise annual reports, we search, match and count the word frequency according to the characteristic words, classify and collect the word frequency of the enterprise digital transformation, obtain the total word frequency and construct an index system for enterprise digital transformation. Referring to general practice, the logarithmic summed word frequency is used to measure the degree of enterprise digital transformation and as the explained variable dt in this study.

#### 3.1.2. Independent Variables

The core explanatory variable is public environmental attention (pec). Drawing lessons from previous research, Baidu keyword search is used as the proxy variable of public environmental attention [53,54,55]. With the development of the internet, web search data based on recording the behavior of netizens in internet searches can attract the attention of market subjects to specific events in a timely manner and reflect their preferences and behavioral intentions [56]. Environmental concerns are an important embodiment of public environmental preferences. Public environmental concerns play a role in coordinating the consistency of public environmental protection behavior and can, thus, measure the general nature of public environmental participation. Specifically, the keyword of the Baidu search index in this work is “environmental pollution”. Compared with other keywords of environmental issues, “environmental pollution” covers a wider range of environmental aspects, more directly reflecting the overall public attention being paid to increasingly serious environmental problems. According to the search channel, the Baidu Index is divided into the total search index, PC search index and mobile search index, in which the total search index is equal to the weighted sum of the PC search index and mobile search index.

#### 3.1.3. Control Variables

This study focuses on capital structure, enterprise performance, asset scale, regional marketization processes and other factors, which are regarded as control variables in the model. Specific variables are selected as follows: assets-to-liabilities ratio (debt), total asset growth rate (growth), total asset net profit margin (roa), management expense ratio (cost), ownership structure (hshare), enterprise age (age) and enterprise size (size). The control variables at the regional level include industrial structure (secrate), economic level (gdp), provincial marketization level (market) and provincial environmental regulation (law). Among them, the marketization-level data are taken from the “China Marketization Index” compiled by Fan and Wang [57]. The environmental regulation intensity data of the province in which the enterprise is located are based on the research of Liu and He [58]. The proportion of industrial pollution control investment in the secondary industry is used to measure environmental regulation.

#### 3.1.4. Moderator Variable

This study examines the regulatory effect of public environmental attention on the digital transformation of enterprises from the nature of property rights, green image and technological basis. Among them, the property rights information of enterprises comes from the CSMAR database; the green image variable is the dummy variable for whether an enterprise has passed the ISO14001 certification. This study measures the technical basis of the enterprise according to whether it independently applies for invention patents or utility model patents in that year, with enterprise patent information coming from the China Research Data Service Platform (CNRDS). The exploration of the abovementioned possible regulatory effects can provide important policy implications for accelerating the promotion of green innovation strategy.

#### 3.1.5. Data Sources

This study uses listed companies in China’s A-share heavy pollution industry as the research sample. The identification of heavily polluting industries is mainly based on the “Guidelines on Industry Classification of Listed Companies” revised by the China Securities Regulatory Commission in 2012, the “List of Listed Companies’ Environmental Verification Industry Classification and Management” formulated by the Ministry of Environmental Protection in 2008 (Huanban Letter (2008) No. 373) and the “Guidelines on Environmental Information Disclosure of Listed Companies” (Huanban Letter (2010) No. 78), which mainly includes 16 heavily polluting industries such as coal, mining, textiles, tannery, paper, petrochemical, pharmaceuticals, chemicals, metallurgy and thermal power. The data period is the annual data from 2012 to 2020. Except for corporate patent data, all other corporate-level data are obtained from the CSMAR database. Public environmental concern data are obtained from the “index.baidu.com(accessed on October 1, 2022)”, and macro variables of the region in which the enterprises are located are obtained from the National Bureau of Statistics. In view of the fact that the listing status of some enterprises do not meet the research requirements, this study refers to the general practice to treat the total sample as follows: excluding companies operating at a loss for two consecutive years (ST), companies operating at a loss for two consecutive years that have not completed the share reform (SST) and companies operating at a loss for three consecutive years (*ST). In addition, to avoid the influence of outliers, on the basis of excluding the missing samples of the aforementioned variable observations, the sample companies with cash holdings, gearing ratios greater than 1 and return on total assets less than 0 are excluded from this study, and all continuous variables are winsorized at the 1% and 99% quartiles. After the above screening process, this study finally obtains 831 enterprises with a total of 5487 sample observations.

For the definitions and detailed calculations of the relevant variables in this article, please see Table 1.

#### 3.1.6. Descriptive Statistics

Table 2 shows the descriptive statistics of the main variables of the model, including the mean, standard deviation and minimum and maximum values of the observed samples, as well as the 25th, 50th and 75th percentiles. The statistical results show that at present, the dt of the degree of digital transformation of enterprises is weak; the average value is 2.266, which is greater than the 75th percentile value; and the distribution is distributed to the right under the influence of the maximum value, indicating that the digital transformation degree of some enterprises is higher, and that the average level has been improved. Therefore, there is great potential for the digital transformation of most enterprises in China. The average value of the environmental pollution index is slightly less than the median, and it is distributed to the left under the influence of the minimum, indicating that some cities pay less attention to environmental pollution and reduce its average level. Other variables are not described in detail. The correlation matrix of the variables is shown in Table A2.

### 3.2. Model Setting

The explained variable in this study is a nonnegative integer that has the characteristics of counting data. Therefore, a Poisson model or negative binomial model is considered for regression. The difference between the two models lies in the dispersion limitation of the explained variable. According to the preliminary descriptive statistics, the variance of the explained variable dt is 4.954, which is significantly greater than its expected (mean) variance of 2.266. It is judged that there may be “transitional dispersion”, and that a negative binomial counting model should be used. The *p* value of the Vuong test is greater than 0.50, which cannot reject the original hypothesis, indicating that the standard negative binomial regression is better than the zero-inflated negative binomial regression. Therefore, the benchmark model in this work adopts negative binomial regression. To verify the theoretical hypothesis of the impact of public environmental concerns on enterprise digital transformation, the basic regression model designed is as follows:(1)$${\mathrm{dt}}_{it}=\alpha_{1}+\beta_{1}per_{it}+\gamma X+Year+City+u$$
where $per_{it}$ represents the degree of digital transformation of the enterprise, ${\mathrm{dt}}_{it}$ represents the audience’s environmental concern in the city in which the enterprise is located, X represents the control variable in the model, Year is the year fixed effects, City represents the urban fixed effects, and u represents the random disturbance term and follows the standard normal distribution. To avoid any endogeneity problems in the model, the independent variables are all lagged by one period.

Then, to explore the impact of public environmental concerns on the heterogeneity of enterprises with different characteristics, this study adds the interaction term of regulatory variables on the basis of Model (1). The specific model is as follows:(2)$${\mathrm{dt}}_{it}=\alpha_{2}+\beta_{2}per_{it}+\beta_{3}interact_{it}^{*}+\beta_{4}interact_{it}^{*}\times per_{it}+\gamma^{\prime }X+Year+City+u^{\prime }$$

Here, $interact_{it}^{*}$ represents the moderator variables selected, including *soe, iso* and *patent*. To facilitate the interpretation of the coefficient of the interaction term, this study adopts decentralization for continuous variables before introducing the interaction term.

## 4. Results and Discussion

### 4.1. Benchmark Regression

Models (1)–(4) of Table 3 examine the relationship between public environmental concerns and the digital transformation of enterprises in heavily polluting industries, showing the regression results of the gradual addition of control variables and fixed industry, year and regional effects under the full sample, respectively, using standard negative binomial regression (Nbreg). From the regression results, in Models (1)–(4), the impact coefficients of public environmental concern on enterprise digital transformation are positive and pass the 5% significance test. After controlling for other variables and fixed effects, Model (4) passes the 1% significance test, indicating that public environmental concerns play a significant role in effectively promoting the digital transformation of heavily polluting enterprises. Therefore, hypothesis H1 of this study is verified. As an informal environmental regulation, public environmental concerns have strengthened the constraints placed on enterprises to fulfill their environmental protection responsibilities through digital transformation and stimulate the digital transformation power of heavily polluting enterprises.

### 4.2. Robustness Tests

To ensure the robustness of the empirical results, this study mainly conducts three forms of robustness tests.

The first is to replace the estimation method of the model. Table 4, Models (1)–(3), report the results of the use of Poisson regression (Passion), zero-inflated negative binomial regression (Zinb) and zero-inflated Poisson regression (Zip), respectively. The impact coefficient of public environmental concern on the digital transformation of heavily polluting enterprises is significantly positive at the 1% level, which is consistent with the previous conclusions and, to some extent, reduces the bias caused by the model setting.

The second is to adjust the measurement indicators of the explained variables, drawing on the ideas of Yuan Chun et al. (2021), based on the “Management Discussion and Analysis” (MD & A) section of the annual reports of listed companies, using the sum of the frequency of words related to enterprise digitalization divided by the length of the MD&A segment in the annual report to measure the degree of digitization of microenterprises (mda). Listed companies usually describe and disclose their business situation and development plan in the MD & A part of their reports. The evaluation related to enterprise digitalization in this part can focus more on reflecting the actual degree of enterprise digitalization transformation. The larger the value of the mda index, the higher the degree of enterprise digitalization transformation. Model (1) in Table 5 reports the regression results of replacing the explained variable with mda. Public environmental concerns still have a positive impact on enterprise digital transformation, passing the 1% significance test. To a certain extent, such concerns weaken the error caused by the selection of measurement methods for enterprise digital transformation.

The third is to replace the independent variable with other indicators to measure public environmental concerns. Considering that the Baidu Index records the search volume of mobile phones and PCs, and that there may be some heterogeneity when netizens use mobile phones and computers to conduct online searches, this study further uses the results of the Baidu Index on mobile phones and PCs to measure the audience’s environmental attention. Models (2) and (3) of Table 5 report their regression results, which show that both environmental attention on mobile terminals and environmental attention on PCs have a positive impact on the digital transformation of enterprises. In addition, this study also selects “smog” as the Baidu search keyword. For the public, smog draws a relatively strong environmental perception. The regression results of public environmental concern (smog) and the conclusion of this study have not changed. Finally, this study uses the frequency of the occurrence of heavily polluting companies in news media reports as an important dimension of public environmental concerns, which is more influential to companies than is the Baidu Index. The regression results of Model (5) in Table 5 show that public environmental attention plays a significant role in promoting the digital transformation of enterprises, which further confirms that public attention has an impact on corporate behavioral decisions. The above tests all support that the basic conclusions of this study are robust.

### 4.3. Heterogeneity Analysis

#### 4.3.1. Property Rights

This study tests the heterogeneity of state-owned and nonstate-owned enterprises by adding a fictitious variable (soe) of property rights nature and its interaction term with the public’s environmental concern in the model. From Table 6, using Models (1) and (2) of zero inflation negative binomial regression and zero inflation Poisson regression, respectively, the interaction coefficient is found to be significantly negative at the 1% level, indicating that compared with state-owned enterprises, for non-state-owned enterprises, when faced with the public’s concern for the environment, the degree of digital transformation is stronger. A possible explanation for this is that non-state-owned enterprises face greater competitive pressure and are more motivated to cope with external public environmental concerns through digital transformation. However, when faced with the informal regulatory pressure of environmental concern, state-owned enterprises may prefer to directly solve the policy burden by increasing environmental protection investment or pollutant discharge fees [59,60], because the pressure faced by these enterprises to generate economic benefits is relatively small. Thus, hypothesis H2 is verified.

#### 4.3.2. Green Image

Models (1) and (2) in Table 7 report the regression results using corporate green image as the adjusting variable. The coefficient of the interaction items is significantly positive at the 10% level, indicating that the public’s environmental concerns play a stronger role in promoting the digital transformation of enterprises with green image certification compared to those enterprises without such certification. A possible explanation for this is that the enterprises that have passed the green certification have strong environmental protection, and to further maintain their green image, they are more sensitive to the public’s environmental concerns and more willing to realize their environmental protection through digital transformation [61]. Thus, hypothesis H3 is verified.

#### 4.3.3. Technical Basis

Models (1) and (2) in Table 8 report the regression results using the dummy variable of enterprise technological foundation as a moderating variable. The coefficients of the interaction terms are all significantly positive at the 1% level, indicating that firms with a better technological base are more inclined to strengthen their informal environmental regulations, such as by responding to public environmental concerns through enterprise digitalization. A possible explanation for this is that on the one hand, high-tech enterprises are more willing to innovate in technology. With the vigorous development of the digital economy, digital transformation is an important direction for the technological innovation of current enterprises. On the other hand, the technological foundation of high-tech enterprises is better than that of other enterprises, which is more conducive to their implementation of digital transformation. Thus, hypothesis H4 is verified.

### 4.4. Further Analysis

Based on the structural layering of enterprise digital transformation, this study further explores the impact of public environmental concerns on specific types of technologies in the digital transformation of enterprises in heavily polluting industries. Table 9 reports the impact of public environmental concerns on four types of digital transformation technologies: artificial intelligence technology (AIT), cloud computing technology (CCT), big data technology (BDT) and digital technology applications(DTA). It should be noted that when examining the digital transformation of blockchain (Blockchain), there is an insufficient effective sample size, and considering that heavily polluting industries involve less blockchain technology, this article does not examine the different types of blockchain technology separately. According to the results of Models (1)–(4), public environmental attention has a stronger effect on promoting digital transformation, such as enterprise digital technology application, and passes the 1% significance test. This finding shows that for heavily polluting enterprises, the development of digital technology that is more inclined to practical application is the main direction of the digital transformation of enterprises, as well as an important way to implement informal environmental regulations such as public attention. Heavy-polluting enterprises come mainly from traditional energy development industries such as coal, mining and metallurgy. To cope with the pressure of industry competition and environmental performance, the digital transformation of heavily polluting enterprises focuses more on applying digital technology to actual production management [31,62,63] than on the development of cutting-edge digital technologies.

## 5. Discussion

Some studies have explored the impact of public environmental concerns on corporate environmental behavior [24,29]. Most studies have concluded that public environmental concerns can increase corporate environmental investment and promote green technology innovation. In the context of China’s vigorous development of the digital economy, digital transformation has become a new path for enterprises to achieve energy conservation and emission reduction due to its potential for promoting production structure optimization and technological progress. We hope to expand our research on whether public environmental concerns affect the digital transformation of enterprises, especially in heavily polluting industries. Our research conclusion confirms that public environmental concern can promote the digital transformation of enterprises in heavily polluting industries. From the perspective of the purpose of enterprise behavior decision making, this is consistent with the research results of Liao [29] and Peng et al. [24].

Considering the differences in enterprise characteristics, this study further reveals the heterogeneity of the impact of the enterprise nature, green image and technology foundation on the digital transformation of enterprises in terms of public environmental concerns. The results show that for nonstate-owned enterprises, enterprises with a green image and enterprises with a high technological foundation, the public’s attention to the environment can significantly promote their digital transformation. Considering that digital transformation includes different types of technological transformation, our research shows that enterprises in heavily polluting industries are more inclined to choose digital technology applications for such digital transformation. A possible explanation is that for heavily polluting enterprises, the digital transformation of development biased toward practical application can be applied to actual production management in the short term, which can not only improve the production and operation efficiency of enterprises but also reduce the pollutants generated by enterprises in the production process [31,62].

These explorations lay a foundation for future research. This study discusses the impact of public environmental concerns on the digital transformation of enterprises, and it is more representative and practical to use as the research object enterprises in heavily polluting industries. However, due to the limitation of research topics and space, the exploration of the specific application of digital transformation in enterprises is not deep enough, and the relevance between enterprises within the industry has not been considered. Future research can conduct in-depth investigations on representative enterprises to provide empirical evidence of the impact of environmental concerns on enterprise digital technology applications. In addition, considering the communication effect of digital transformation through enterprise supply chain relations also leaves room for further research.

## 6. Conclusions and Policy Recommendations

### 6.1. Conclusions

At present, China adheres to the governance concept of “government responsibility, social coordination and public participation”. However, the government often faces the problem of insufficient supervision, which requires public participation in environmental governance. This study uses the company data of heavily polluting industries in China’s A-share market from 2012 to 2020 and Baidu Index data and uses a negative binomial model to analyze the impact of public environmental concerns on the digital transformation of heavily polluting enterprises. The results show that (1) public environmental concern can significantly promote the digital transformation of heavily polluting enterprises, and the research results are robust; (2) the heterogeneity analysis indicates that public environmental concerns play a stronger role in promoting the digital transformation of non-state-owned enterprises, green image enterprises and high-tech enterprises compared to other types of enterprises; and (3) based on the structural and hierarchical perspective of enterprise digital transformation, public environmental concerns significantly promote the application of digital technology among enterprises.

### 6.2. Policy Recommendations

This study is helpful for understanding the role of public participation in environmental governance in improving environmental quality and has certain enlightenment significance for enterprises in improving their environmental awareness, shaping their green image, and consciously carrying out green innovation. The public should constantly improve its awareness of environmental protection, actively participate in the ranks of environmental governance, play a role of environmental protection supervisor and make up for the limitations of formal systems in the process of environmental governance. Enterprises, in the process of seeking maximum economic benefits, should shoulder their green social responsibility under the construction of an ecological civilization, establish and strengthen the concept of green development, form a good green image, carry out digital transformation driven by green innovation and exhibit the goal and determination of enterprise environmental governance to society and the public with practical actions. The government should promote public environmental protection policies with legal systems, improve environmental regulation and environmental information disclosure systems, and introduce relevant policies to promote the digital transformation of enterprises. At the same time, such transformation provides more convenient channels for public environmental participation and constantly improves public environmental participation and attention.

## Figures and Tables

**Table 1 ijerph-20-00203-t001:** Variable definitions.

	Variable	Indicator	Definition
Dependent variable	dt	Degree of digital transformation	Ln (Word frequency related to digital transformation in annual report)
	mda	Degree of digital transformation (%)	Percentage of word frequency related to digital transformation in the “Management’s Discussion and Analysis” section of the annual report
Independent variable	pec	Public environmental concern	Ln (“Environmental Pollution” Index + 1)
	smog	“Smog” Baidu Index	ln (“Smog” Index + 1)
	news	Media attention	ln (total number of online financial news headlines appearing for the company + 1)
Control variables	debt	Assets-to-liabilities ratio	Total liabilities/total assets
	growth	Growth rate of total assets	Annual growth rate of total assets
	roa	Net profit margin of total assets	Net profit/total asset balance
	cost	Management expense rate	Administrative expenses/operating income
	hshare	Ownership structure	Shareholding ratio of the largest shareholder
	age	Enterprise age	Year the company went public
	size	Enterprise scale	Ln (total assets)
	secrate	Regional industrial structure	Proportion of secondary industry
	gdp	Regional economic level	Ln (Urban GDP per capita)
	market	Marketization level	Total score of the provincial marketization process
	law	Environmental regulation	Provincial Environmental Regulation Intensity Index
Moderator variables	soe	Property right nature	1 = nature of equity is state-owned enterprise; otherwise = 0
	iso	Green image	1 = ISO14001 certification; 0 = otherwise
	patent	Technical foundation	1 = independent application for invention patent or utility model patent in the current year; 0 = otherwise

**Table 2 ijerph-20-00203-t002:** Descriptive statistics.

Variable	N	Mean	Standard Deviation	Min.	Max.	P25	P50	P75
dt	5487	2.266	4.954	0.000	30.000	0.000	0.000	2.000
mda	5487	0.017	0.041	0.000	0.258	0.000	0.000	0.015
pec	5487	3.781	0.929	1.345	5.080	3.171	3.943	4.561
smog	5487	4.174	1.436	0.485	6.900	3.458	4.473	5.113
news	5487	4.439	1.150	0.000	6.596	3.951	4.585	5.165
debt	5487	0.396	0.197	0.046	0.822	0.233	0.388	0.549
growth	5487	0.166	0.280	−0.203	1.684	0.020	0.092	0.211
roa	5487	0.048	0.049	−0.106	0.199	0.016	0.042	0.075
cost	5487	0.080	0.051	0.008	0.272	0.043	0.070	0.103
hshare	5487	0.363	0.151	0.091	0.794	0.250	0.345	0.464
age	5487	10.755	7.046	1.000	26.000	4.000	10.000	17.000
size	5487	22.253	1.354	20.015	26.315	21.270	22.012	23.031
secrate	5487	0.443	0.108	0.183	0.670	0.386	0.459	0.517
gdp	5487	15.682	0.529	14.408	16.631	15.315	15.743	16.101
market	5487	7.934	1.935	2.920	10.960	6.550	7.940	9.670
law	5487	0.002	0.002	0.000	0.009	0.001	0.002	0.003
soe	5487	0.387	0.487	0.000	1.000	0.000	0.000	1.000
iso	5487	0.249	0.433	0.000	1.000	0.000	0.000	0.000
patent	5487	0.497	0.500	0.000	1.000	0.000	0.000	1.000

**Table 3 ijerph-20-00203-t003:** Benchmark regression.

	(1)	(2)	(3)	(4)
	Nbreg	Nbreg	Nbreg	Nbreg
Variable	dt	dt	dt	dt
pec	0.236 ***	0.198 ***	0.122 **	0.173 ***
	(8.00)	(6.64)	(2.25)	(3.15)
debt		−0.454 **	−0.233	−0.047
		(−2.14)	(−1.07)	(−0.22)
growth		0.227 **	0.233 **	0.240 **
		(1.98)	(2.11)	(2.23)
roa		0.045	0.148	1.158
		(0.07)	(0.20)	(1.58)
cost		−0.319	0.599	0.260
		(−0.45)	(0.86)	(0.38)
hshare		−0.161	0.039	0.072
		(−0.80)	(0.19)	(0.36)
age		−0.000	−0.002	−0.003
		(−0.03)	(−0.35)	(−0.60)
size		0.234 ***	0.249 ***	0.211 ***
		(7.17)	(7.51)	(6.49)
secrate			−2.709 ***	−1.381 ***
			(−5.60)	(−2.66)
gdp			−0.041	−0.179 *
			(−0.44)	(−1.92)
market			0.560 ***	−0.141
			(10.47)	(−1.50)
law			53.362 **	−28.640
			(1.99)	(−0.90)
Constant	−0.101	−6.036 ***	−10.321 ***	−2.609
	(−0.86)	(−8.20)	(−7.06)	(−1.62)
lnalpha	1.376 ***	1.308 ***	1.129***	1.071 ***
	(49.30)	(44.91)	(35.52)	(33.30)
Industry fixed effects	No	Yes	Yes	Yes
Year fixed effects	No	No	No	Yes
Region fixed effects	No	No	Yes	Yes
N	5487	5487	5487	5487
r2_p	0.003	0.013	0.040	0.049
chi2	64.026	282.241	917.553	1096.204

Notes: robust standard errors in parentheses. *** *p* < 0.01, ** *p* < 0.05, * *p* < 0.1.

**Table 4 ijerph-20-00203-t004:** Robustness test: replacement model estimation method.

	(1)	(2)	(3)
	Poisson	Zinb	Zip
Variable	dt	dt	dt
pec	0.201 ***	0.173 ***	0.092 ***
	(3.60)	(3.00)	(4.36)
Constant	−2.248	−2.609	−0.472
	(−1.37)	(−1.64)	(−0.81)
lnalpha		1.071 ***	
		(35.54)	
Control variables	Yes	Yes	Yes
Industry fixed effects	Yes	Yes	Yes
Year fixed effects	Yes	Yes	Yes
Region fixed effects	Yes	Yes	Yes
N	5487	5487	5487
r2_p	0.184		
chi2	1046.452	950.301	2957.855

Notes: robust standard errors in parentheses. *** *p* < 0.01.

**Table 5 ijerph-20-00203-t005:** Robustness tests: adjusted measurement method.

	(1)	(2)	(3)	(4)	(5)
	Digital Word Frequency Ratio	PC-Side “Environmental Pollution” Baidu Index	Mobile Phone “Environmental Pollution” Baidu Index	Total “Smog” Baidu Index	News Media Attention
Variable	mda	dt	dt	dt	dt
pec	0.286 ***				
	(4.91)				
computer		0.157 **			
		(2.49)			
phone			0.143 ***		
			(3.05)		
smog				0.180 ***	
				(2.71)	
news					0.060 **
					(2.49)
Constant	−2.063	−2.771 *	−2.644	−2.658	−4.677 ***
	(−1.31)	(−1.65)	(−1.61)	(−1.63)	(−3.19)
lnalpha	−30.165	1.072 ***	1.071 ***	1.071 ***	1.073 ***
		(33.39)	(33.30)	(33.24)	(33.47)
Control variable	Yes	Yes	Yes	Yes	Yes
Industry fixed effects	Yes	Yes	Yes	Yes	Yes
Year fixed effects	Yes	Yes	Yes	Yes	Yes
Region fixed effects	Yes	Yes	Yes	Yes	Yes
N	5487	5487	5487	5487	5487
r2_p	0.087	0.049	0.049	0.049	0.049
chi2	1365.881	1091.238	1111.934	1081.071	1081.898

Notes: robust standard errors in parentheses. *** *p* < 0.01, ** *p* < 0.05, * *p* < 0.1.

**Table 6 ijerph-20-00203-t006:** Heterogeneity analysis based on property rights nature.

	(1)	(2)
	Nbreg	Poisson
Variable	dt	dt
pec	0.245 ***	0.258 ***
	(4.02)	(4.34)
soe×pec	−0.201 ***	−0.154 ***
	(−3.35)	(−2.63)
soe	0.232	0.107
	(0.97)	(0.46)
Constant	−2.727 *	−2.239
	(−1.69)	(−1.41)
lnalpha	1.050 ***	
	(32.09)	
Control variable	Yes	Yes
Industry fixed effects	Yes	Yes
Year fixed effects	Yes	Yes
Region fixed effects	Yes	Yes
N	5487	5487
r2_p	0.052	0.193
chi2	1149.146	1097.761

Notes: robust standard errors in parentheses. *** *p* < 0.01, * *p* < 0.1.

**Table 7 ijerph-20-00203-t007:** Heterogeneity analysis based on green image.

	(1)	(2)
	Nbreg	Poisson
Variable	dt	dt
pec	0.129 **	0.171 ***
	(2.23)	(3.03)
iso × pec	0.134 *	0.125 *
	(1.91)	(1.66)
iso	−0.629 **	−0.623 **
	(−2.25)	(−2.06)
Constant	−2.908 *	−2.180
	(−1.81)	(−1.33)
lnalpha	1.069 ***	
	(33.21)	
Control variable	Yes	Yes
Industry fixed effects	Yes	Yes
Year fixed effects	Yes	Yes
Region fixed effects	Yes	Yes
N	5487	5487
r2_p	0.049	0.185
chi2	1120.366	1058.567

Notes: robust standard errors in parentheses. *** *p* < 0.01, ** *p* < 0.05, * *p* < 0.1.

**Table 8 ijerph-20-00203-t008:** Heterogeneity analysis based on technical basis.

	(1)	(2)
	Nbreg	Poisson
Variable	dt	dt
pec	0.107 *	0.135 **
	(1.80)	(2.23)
patent×pec	0.175 ***	0.194 ***
	(2.99)	(3.18)
patent	−0.746 ***	−0.826 ***
	(−3.09)	(−3.31)
Constant	−2.535	−2.248
	(−1.56)	(−1.38)
lnalpha	1.067 ***	
	(33.23)	
Control variable	Yes	Yes
Industry fixed effects	Yes	Yes
Year fixed effects	Yes	Yes
Region fixed effects	Yes	Yes
N	5487	5487
r2_p	0.050	0.186
chi2	1110.510	1057.222

Notes: robust standard errors in parentheses. *** *p* < 0.01, ** *p* < 0.05, * *p* < 0.1.

**Table 9 ijerph-20-00203-t009:** Specific types of digital transformation technologies.

	(1)	(2)	(3)	(4)
Variable	AIT	CCT	BDT	DTA
pec	0.099	−0.093	−0.098	0.268 ***
	(0.83)	(−0.79)	(−1.21)	(4.07)
Constant	−18.414 ***	−1.447	−14.564 ***	−2.947
	(−5.12)	(−0.36)	(−5.72)	(−1.53)
lnalpha	1.963 ***	2.237 ***	1.371 ***	1.428 ***
	(20.06)	(30.69)	(20.83)	(36.08)
Control variable	Yes	Yes	Yes	Yes
Industry fixed effects	Yes	Yes	Yes	Yes
Year fixed effects	Yes	Yes	Yes	Yes
Region fixed effects	Yes	Yes	Yes	Yes
N	5487	5487	5487	5487
r2_p	0.114	0.068	0.090	0.054
chi2	3305.125	8257.780	479.037	6397.887

Notes: robust standard errors in parentheses. *** *p* < 0.01.

## Data Availability

Publicly available datasets were analyzed in this study. This data can be found at: https://www.gtarsc.com/.

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
