# Peer review of "Impact of Public Environmental Concerns on the Digital Transformation of Heavily Polluting Enterprises"

_ijerph, 2022, doi:10.3390/ijerph20010203_

Round 1

Reviewer 1 Report

Ls155-157: The second hypothesis is confusingly expressed. It can be changed to "Compared to non-state-owned enterprises, state-owned enterprises..." or "Compared to state-owned enterprises, non-state-owned enterprises..."

L194: What is "ST,SST,*ST"? When an acronym first appears in the text, an explanation is required.

Ls193-196: It needs to be explained to the reader why the raw data should be filtered according to these principles.

L434: The manuscript is missing the discussion section. It is necessary to insert a new paragraph “discussion” to debate about the resulting data, highlighting what makes this manuscript different compared to the previous research results. In my opinion, "4.4" is what I would be more interested in compared to the other results.

L370: If possible, it is recommended to use instrumental variables to deal with endogeneity problem.

Reviewer 2 Report

Impact of Public Environmental Concerns on the Digital Trans-formation of Heavily Polluting Enterprises

The authors contend that public environmental concern “significantly promoted” the digital transformation of heavily polluting enterprises in China 

The authors seemingly confuse correlation with causality. It is easy to assume that public concern for pollution increased over the decade studied although this basic data is not reported. In like manner one would assume that many “heavy industries” would report advances in the use of digital tools (digital transformation) over the period.  What is missing is the real link in either theory or data showing how public environmental concern would cause a firm to increase its digital transformation.  This is a fatal flaw in the study.

Would not environmental concern more likely cause a firm to invest in pollution reducing industrial processes and equipment.  The authors operationalize digital transformation with artificial intelligence, block chain cloud computing, big data and digital tech applications.  These may or may not have a direct effect on a firm’s environmental pollution.   

The authors apply a number of regression based analyses to explore the correlation between their two key variables. The first order correlation is reported as 0.103 and is of course significant given a sample size of 5,487 observations. In like manner all the other analyses are statistically significant but produce extremely low levels of explained variation. All the analyses and statistics obscure the missing link between public concern over pollution and an enterprise’s move to utilize more advanced digital technology in general.     

The authors explore 3 hypotheses across the period of 2012 through 2020 for 831 enterprises yielding 5,487 observations. In the introduction heavily polluting industries are identified as steel, petrochemical, cement and chemical industries however the identification of enterprises in the study is unclear. 

Hypothesis 1. Public environmental concerns have significantly promoted the digital transformation of heavily polluting enterprises.

Hypothesis 2. The nature of enterprise property rights plays a negative role in regulating the impact of public environmental concerns on the digital transformation of heavily polluting enterprises.

Hypothesis 3. Corporate green image plays a positive role in regulating the impact of public environmental concerns on the digital transformation of heavily polluting enterprises.

Digital transformation is operationalized from word searches of annual reports.  The words indicated in the notes are Artificial intelligence, business intelligence, image understanding, biometrics, etc. Cloud computing, stream computing, graph computing, memory computing, Internet of Things, etc. Blockchain, digital currency, distributed computing, differential privacy technology, etc. Big data, data mining, text mining, data visualization, etc. Mobile internet, e-commerce, mobile payment, B2B, B2C, C2C, O2O, etc.

The authors could have ( should have) searched for investments in pollution reducing investments or technology rather than the general digital transformation taking place in enterprises large and small polluting and non-polluting over the time period examined. 

Public environmental concerns which the authors call “public attention” is operationalized by counting internet word searches for “environmental pollution.”  An internet search may be a reasonable surrogate for public concern, but it does not indicate public pronouncements against polluting enterprises that might cause them to change their behavior. (I suspect the level of control over the internet reported in China may affect public outcry against state enterprises.)  Public media sources would be a better source of a measure of public attention especially if specifically related to the enterprises studied.     

Reviewer 3 Report

Overall, there is a lot to like about this paper.  It represents an ambitious effort to understand how public environmental concerns translate into corporate policy decisions in the context of the Chinese political system.  This is an intriguing project and analysis. 

I would like to see a bit more robustness checking though.  These involve both the key independent variable and key dependent variable. 

First, concerning the search frequency independent variable, any article using search variables as a measure must recognize that there are an enormous number of different possible search variables.  As such, I believe that disclosure of how many different search terms were examined before the search term used was settled upon is a must -- this disclosure can help minimize the risk of data mining to find spurious correlations. 

Second, concerning the digitization dependent variable, a paragraph could potentially be added further laying out the case for why digitization is closely related to environmental protection.  There is already some discussion of this in the paper, but I think it should be expanded a bit further. 

Reviewer 4 Report

- Overall interesting. The impact of public concern on public companies is well known in the West, and the conventional wisdom would be that the same is true in China. This work seems to confirm that and to suggest that public influence is less impactful in state-owned enterprises. We see that in the West as well. - English is good. Some punctuation and grammar editing needed in the Data & Methodology section (missing punctuation, excess or missing words, etc.). - The discussion of the SMOG variable (lines 279-283) is confusing. How can both statements about attention to pollution be true when they are contradictory? - Several terms are marked with an asterisk (*) indicating that a further explanation is provided, but the explanation never appears. - Why is Table A1 included in the work but never mentioned in the text of the manuscript?

Reviewer 5 Report

The article has no features or major problems, and the following contents need to be modified:

1. appropriately compress the research background;

2. Data cycle shall be described;

3. The tables in the paper are not standardized enough;

4.In the results and discussion section, there is no discussion.

Reviewer 6 Report

The present manuscript uses regression models to test 4 hypothesis surrounding how public environmental concerns impact the public facing behaviour of heavily polluting enterprises. The manuscript is of good scientific and editorial quality. However some key terminologies are poorly defined, and some text is superfluous. The conclusion in particular is well written. Detailed comments below.

Naming. Some of the variable names are too long and unwieldy; please simplify (e.g. i.IsPAssISO14001#c.pec). When shortening terms like negative binomial regression to nbreg, please define the term at first use.

Key concepts such as heavily polluting enterprises, or state owned enterprises, are not sufficiently defined. The manuscript is often confusing because it discusses many broad concepts (governance systems, “mandatory purpose” in line 140) without sufficiently defining these terms. At the same time, the introduction could be made more concise, because there are lots of interesting details of little consequence to the discussion.

Smog and haze are used interchangeably, please choose one term.

Tables 3 onwards: Please provide table figures and definition of asterisks.

Line 146: why the emphasis on punishment, rather than influence in general?

196: Which variables?
198: The data came from, or the data comes from

219: what notes?

Incomplete sentence line 261

Round 2

Reviewer 2 Report

Manuscript revision completely changes the focus of the study is is now an interesting and publishable study.